# The Role of Macrophage Iron Overload and Ferroptosis in Atherosclerosis

**DOI:** 10.3390/biom12111702

**Published:** 2022-11-18

**Authors:** Jiedong Ma, Hongqi Zhang, Yufei Chen, Xiaojin Liu, Jiamin Tian, Wei Shen

**Affiliations:** 1Department of Cardiology, Huashan Hospital, Fudan University, Shanghai 200040, China; 2Department of Anatomy, Histology and Embryology, Shanghai Medical College, Fudan University, Shanghai 200032, China

**Keywords:** ferroptosis, atherosclerosis, macrophages, iron homeostasis, iron overload

## Abstract

Ferroptosis is a new type of cell death caused by iron-dependent lipid peroxidation. In recent years, it has been found that ferroptosis can promote the progression of atherosclerosis (AS). Macrophages have been proven to play multiple roles in the occurrence and development of AS. Iron is a necessary mineral that participates in different functions of macrophages under physiological conditions. But iron overload and ferroptosis in macrophages may promote the progression of AS. Herein, we summarize the role of iron overload and ferroptosis in macrophages in AS from the perspective of iron metabolism, and iron overload and ferroptosis are significant contributors to AS development.

## 1. Introduction

Cardiovascular diseases (CVDs) pose a serious threat to health, especially in the elderly. Atherosclerotic cardiovascular disease (ASCVD) is a particularly notable form of CVD. The 2021 European Society of Cardiology (ESC) Guidelines pointed out that ASCVD was still the main cause of morbidity and mortality of all CVDs [1]. The prevalence of risk factors for ASCVD such as smoking, obesity, hypertension, and diabetes continues to increase. Thus, the importance of preventing ASCVD remains undisputed [2]. Studying the mechanisms underlying the development of ASCVD would be helpful in developing new ways to prevent and treat this disease. Atherosclerosis (AS) is known to be the key pathological mechanism of ASCVD but features a highly complex mechanism.

Macrophages, as effector cells of protective immunity, play multiply roles in AS. The number and phenotype of macrophages in plaques are associated with the progression and remission of AS [3]. In the initial stage of AS, macrophages combine with lipopolysaccharide (LPS) to modify internal lipoprotein, promote cholesterol accumulation and intensify the inflammatory reaction. Then, macrophages phagocytize and accumulate lipid peroxide (LPO), producing foam cells [4]. Foam cells facilitate the accumulation of lipids in plaques and promote the development of AS plaques. Macrophages also exacerbate inflammatory responses by promoting the secretion of cytokines, chemokines and reactive oxygen species (ROS). Furthermore, macrophages contribute to the death of smooth muscle cells and reduce the stability of plaques. During the end stage of AS, plaques become complicated and fragile. The process of apoptosis in macrophages is aggravated in an inflammatory environment. If apoptotic cells are not eliminated in a timely manner, necrosis will occur and necrotic cores will begin to form.

Apoptosis in a small number of macrophages can exert a protective effect during the early stages of AS. However, macrophages apoptosis can also be deleterious at the advanced stage of AS [5]. Apart from apoptosis, other forms of macrophage death may also exert influence on AS such as macrophages made for about half of the dead cell population in advanced AS plaques (Table 1). Studies have found that nicotine can promote the progression of AS by affecting pyroptosis in macrophages [6]. The process of necrosis in macrophages may induce inflammation and expand the necrotic core of AS plaques [7]. In macrophages, autophagy is one of the protective factors of AS and can delay the progression of AS [8]. In addition, the process of ferroptosis in macrophages can also have an important effect on AS. Previous researchers reported that macrophage ferroptosis played a key role in the instability of AS plaques [9]. Consequently, regulating macrophage ferroptosis may be a potential important way to stabilize plaques and inhibit the progression of AS. Iron overload is one key step in the occurrence of ferroptosis. In addition, iron overload can promote the development of AS. Therefore, we searched PubMed and Web of Science using the following terms “ferroptosis” (topic) and “atherosclerosis” (topic), plus “ferroptosis” (topic) and “macrophage” (topic), plus “iron” (topic) and “macrophage” (topic). Non-English literature and non-full text were excluded. In this review, we discuss the effect of iron overload and ferroptosis in macrophage on AS from the perspective of iron metabolism.

## 2. Ferroptosis

### 2.1. Definition of Ferroptosis

Dixon [10] found an iron-dependent non-apoptotic form of cell death called ferroptosis which was characterized by lipid peroxidation. Disturbance of iron homeostasis leads to iron overload; subsequently, an excess of ferrous iron (Fe^2+^) causes lipid peroxidation through the Fenton reaction and leads to redox imbalance. Large amounts of polyunsaturated fatty acids (PUFAs) are then oxidized, ultimately leading to ferroptosis. Unlike apoptosis, the specific morphology of ferroptosis under microscopy presents with membrane destabilization and disruption, cytoplasmic and organelle swelling, mitochondrial atrophy, increased mitochondrial membrane density and the reduction or disappearance of the mitochondrial crest but without any change in nuclear morphology [11]. An overview of ferroptosis is shown in Figure 1.

Extracellular iron is taken up by cells via TFR1 and iron is excreted out of cells through FPN. Iron overload can occur when iron deposition is caused by an imbalance in intracellular iron homeostasis. Iron overload can lead to lipid peroxidation by Fenton reaction and promote ferroptosis. Furthermore, LOXs, POR, NOXs can also produce LPO through enzymatic reactions. There are three pathways that cells eliminate LPO: the GPX4-glutathione axis, the FSP1-CoQ10 axis and the GCH1-BH4 axis. Abbreviations: Fe^3+^, ferric iron; Fe^2+^, ferrous iron; TFR1, transferrin receptor 1; FPN, ferroportin; DFO, deferoxamine; PUFA, polyunsaturated fatty acid; ACSL4, Acyl-coenzyme A synthetase long-chain family member 4; CoA, coenzyme A; LPCAT3, lysophosphatidylcholine acyltransferase 3; PL, phospholipids; LOXs, lipoxygenases; POR, cytochrome P450 oxidoreductase; NOXs, reduced nicotinamide adenine dinucleotide phosphate oxidase; Glu, glutamic acid; Cys, cysteine; γ-Gcs, γ-glutamylcysteine; Gly, glycine; GSH, reduced glutathione; GSSH, oxidized glutathione; GSR, glutathione S-reductase; NADPH, reduced form of nicotinamide-adenine dinucleotide phosphate; GPX4, glutathione peroxidase 4; GCH1, GTP-cyclohydrolase1; BH4, tetrahydrobiopterin; CoQ10H2, ubiquinol; CoQ10, ubiquinone; FSP1, Ferroptosis suppressor protein 1; FINs, ferroptosis-inducing compounds; RSL3, Ras Selective Lethal 3.

### 2.2. Mechanism and Regulatory Factors of Ferroptosis

Iron overload and lipid peroxidation are key steps in the occurrence of ferroptosis. Extracellular iron is taken up by cells via transferrin receptor 1 (TFR1). In cells, iron is mainly stored as complexes with ferritin (FT) or stored in a labile iron pool (LIP). Excessive cellular iron is excreted by an iron transporter protein called ferroportin (FPN). Intracellular iron homeostasis normally remains stable under physiological conditions. However, when iron metabolism is out of balance, the excessive active form of Fe in cells may lead to iron overload.

Iron overload causes lipid peroxidation; this results in excessive amounts of Fe^2+^ which can then produce ROS via the Fenton reaction. ROS tends to attack carbon-carbon double bonds in lipids, especially PUFAs, which are an important component of the cell membrane. Lipid peroxidation may ultimately lead to ferroptosis. Of the various forms of PUFAs, arachidonic acid and adrenic acid have been shown to be important factors in ferroptosis. The process in which Fe^2+^ produces ROS and eventually oxidizes PUFAs is referred to as the non-enzymatic oxidation pathway. Enzymatic oxidation pathways also play a key role in lipid peroxidation. For example, lipoxygenases (LOXs) [12], cytochrome P450 oxidoreductase (POR) [13] and reduced nicotinamide adenine dinucleotide phosphate oxidase (NOXs) can produce LPO via enzymatic reactions.

The occurrence of ferroptosis can be influenced by many elements; in this review, we summarize the key factors that are associated with iron metabolism. For example, ammonium ferric citrate, nuclear receptor coactivator 4 (NCOA4), siramesine and lapatinib can all induce ferroptosis, while ferrostatin-1 (Fer-1), heat shock protein 27 and deferoxamine (DFO) can inhibit ferroptosis (Table 2). However, the specific mechanisms of some of these regulators in macrophages are still unclear and need to be investigated further [14,15,16,17,18,19,20,21,22,23,24].

### 2.3. Anti-Ferroptosis Systems in Cells

Three systems are involved in ferroptosis: the GPX4-GSH axis [25], the FSP1-CoQ10 axis [26] and the GCH1-BH4 axis [27].

The GPX4-GSH axis is the main pathway in the anti-ferroptosis system. Glutathione peroxidase 4 (GPX4) utilizes glutathione (GSH) as a reducing substance to counteract lipid peroxidation. Cystine is an important synthetic material for GSH. Extracellular cystine is transported into cells through system Xc-, which is composed of subunit solute carrier family 7 member 11 (SLC7A11) and solute carrier family 3 member 2 (SLC3A2). Ferroptosis-inducing compounds 1 (FINs1), such as elastin, can inhibit system Xc-. FINs2, such as Ras Selective Lethal 3 (RSL3), can directly inhibit GPX4 and promote the overwhelming generation of ROS. Nicotinamide-adenine dinucleotide phosphate (NADPH) is required for recycling GSH in the GPX4-GSH axis [28]. Any abnormality in the GPX4-GSH axis may promote the overwhelming generation of ROS that results in lipid peroxidation, including GPX4 gene knockout [29], the inhibition of system Xc- or GPX4, or deficiencies in GSH, cysteine or NADPH. Cysteine deficiency, for example, can promote ferroptosis because cysteine is a rate-limiting substrate for GSH production and the subsequent glutamate buildup raises ROS. Fortunately, when the GPX4-GSH axis is inhibited, the expression of prominin 2 is enhanced and prominin 2 promotes the formation of ferritin-containing multivesicular bodies and exosomes, which can then inhibit ferroptosis by stimulating the export of iron [30].

Ferroptosis suppressor protein 1 (FSP1), which used to be referred to as apoptosis-inducing factor mitochondrial 2 (AIFM2), is a powerful inhibitor of ferroptosis. There are two domains in FSP1, N-myristoylation and a flavoprotein oxidoreductase domain. N-myristoylation attracts FSP1 to the plasma membrane. Once recruited, FSP1 regenerates reductive ubiquinol (CoQ10H2) by oxidizing ubiquinone (CoQ10) with NADPH. CoQ10H2 plays a key role in suppressing lipid peroxides generation and ferroptosis. FSP1 has been found in cell membranes but not in the mitochondria. Dihydroorotate dehydrogenase (DHODH) is found inside mitochondria and functions in a similar manner as FSP1. DHODH catalyzes dihydroorotate to orotate and protects against ferroptosis, independent of cell membrane FSP1, cytoplasmic GPx4 and mitochondrial GPx4 [31]. The DHODH axis and the GPX4 axis are two parallel pathways in mitochondria. The sensitivity of one pathway is enhanced when the other pathway is inhibited. DHODH inactivation may result in mitochondrial lipid peroxidation and inflammation when GPx4 expression is suppressed [32].

GTP-cyclohydrolase1 (GCH1) produces tetrahydrobiopterin (BH4), which counteracts lipid peroxidation in two ways. On the one hand, BH4 itself captures peroxyl radicals produced from lipids and can reduce oxidized lipids. On the other hand, BH4 can help to produce CoQ10H2 [33].

### 2.4. Atherosclerosis and Ferroptosis

During the early stages of AS, lipid deposition and macrophage aggregation are the main manifestations and can be accompanied by a mild inflammatory response. With the development of AS, the inflammatory reaction is intensified, and lipoprotein deposition becomes more obvious. Plaques form gradually and become increasingly less stable. At the end stage of AS, the stability of the plaques significantly decreases and the plagues begin to rupture; blood vessels show severe stenosis or even complete occlusion; cardiovascular events are also more likely to occur, such as heart failure, myocardial infarction, arrhythmias and even sudden death.

During the development of AS, various forms of cell death, including apoptosis, ferroptosis, efferocytosis and pyroptosis are inevitable. Ferroptosis has been proven to be closely associated with AS. Although the specific mechanism of ferroptosis in AS is not clear, its specific role in AS is worth exploring. The key characteristics of ferroptosis include iron overload, lipid peroxidation and inflammatory reactions. In addition, iron deposition, redox imbalance, and inflammation are characteristics of human AS plaques.

Ferroptosis-related molecules and factors are closely associated with AS. For example, the development of AS can be accelerated by the excess of Fe^2+^. Moreover, iron chelation and the limitation of dietary iron can stabilize AS plaques and defend against endothelial impairment. Oxidative stress can also increase the risk of CVDs; ROS can make a significant contribution to the progression of AS plaques [34]. Furthermore, the expression of GPX4 is inversely correlated with AS severity [35]. The inhibition of GPX4 might result in the accumulation of lipid peroxides, ferroptosis and the development of atherosclerotic plaques [36]. Ferroptosis-related LOXs accelerate the development of AS, whereas suppressing 12/15-LOX reduces LDL oxidation and AS [37]. Heat shock protein 27 can down-regulate TFR1 to reduce iron absorption and combat ferroptosis [38] and can also inhibit the growth of foam cells and AS plaques [39]. Fer-1, an inhibitor of ferroptosis, reduces iron uptake and ROS production and can also delay the progression of AS by reducing lipid peroxidation and endothelial dysfunction [20]. Apart from LOXs, two other enzymes, prostaglandin endoperoxide synthase 2 (PTGS2) and acyl-CoA synthetase long-chain family member 4 (ACSL4), also play an important role in phospholipid metabolism and ferroptosis. Additionally, ferroptosis-related proteins PTGS2 and ACSL4 are upregulated as AS progresses. During the early stage of AS, ferroptosis-related proteins may not show significant differences in expression when compared with a control group; thus, cell death may not be a major feature of early plaques. However, as AS develops, ferroptosis occurs gradually, and the expression levels of ferroptosis-related proteins also start to show differences [35].

In addition, many genes are strongly associated with both ferroptosis and AS. *Nrf2* is known to regulate ATP-binding cassette transporter B6 (ABCB6); the downregulation of ABCB6 is known to prevent ferroptosis and slow the development of AS. In addition, *Nrf2* can increase the expression of SLC7A11 and the synthesis of GSH [40], which counteracts iron oxidation and lipid peroxidation. In addition, GSH reduces the inflammatory response and artery protection, thus reducing the progression of AS. *PDSS2* activates *Nrf2* and thus inhibits the formation of atherosclerotic plaque lesions and reduces ferroptosis [41]. *P53* and *ATF3* can prevent the synthesis of GSH by binding to SLC7A11. Furthermore, *P53* can accelerate ferroptosis and the development of AS by activating glutaminase2 [42], thus increasing the hydrolysis of GSH and the generation of ROS. *STAT3* can exert impact on the immunological response, inflammatory response, macrophage polarization and endothelial cell function. The activation of *STAT3* can halt the development of AS [43]. In addition, *STAT3* controls lipid peroxidation and iron metabolism to prevent ferroptosis. *SCD1* can reduce the inflammatory response of macrophages and endothelial cells, acting as a regulator of AS. *SCD1* can also convert saturated fatty acids into monounsaturated fatty acids and prevent ferroptosis [44]. These results mentioned above suggest that ferroptosis plays an important role in AS.

## 3. Macrophages and Iron Homeostasis

### 3.1. Iron Homeostasis in Macrophages

There are two main sources of iron in macrophages. First, macrophages phagocytose senescent red blood cells (RBC) or RBCs produced by intra-plaque hemorrhage and produce heme. In phagosomes, heme oxygenase-1 (HO-1) degrades heme to produce Fe^2+^. Second, extracellular ferric iron (Fe^3+^) combines with transferrin (TF) and can enter macrophages by TFR1. In endosomes, Fe^3+^ is reduced into Fe^2+^ by six-segment transmembrane epithelial antigen of prostate 3 (STEAP3). Subsequently, Fe^2+^ is transported into the cytoplasm through divalent metal transporter 1 (DMT1). In cells, iron is mainly stored in an inactive bound form, mainly as complexes with FT, which is composed of ferritin heavy chain (FTH) or ferritin light chain (FTL). FTH has ferrous oxidase activity that can oxidize Fe^2+^ into Fe^3+^ and store it in FT and protect cells from the oxidation and toxic effects of Fe^2+^. FT combines with Fe^3+^ and can be transported out of the cell by FPN [45]. The remaining intracellular iron is stored in the LIP. Under physiological conditions, the balance of iron metabolism in macrophages can be maintained under the regulation of various regulatory factors. However, under pathological situations, abnormal iron metabolism may cause an excess of the active form of iron. This leads to iron deposits and lipid peroxidation, finally contributing to ferroptosis (Figure 2).

Iron sources of macrophages: Macrophages phagocytose RBC and produce Fe^2+^, or extracellular Fe^3+^ enters into macrophages through TFR and is reduced to Fe^2+^. Then, Fe^2+^ is stored in a LIP. Iron combines with FT and is discharged out of cells by FPN. When iron homeostasis is abnormal, iron deposition in macrophages causes lipid peroxidation, which can finally trigger ferroptosis. Abbreviations: TF, transferrin; DMT1, divalent metal transporter 1; RBC, red blood cells; STEAP3, six-segment transmembrane epithelial antigen of prostate 3; HO-1, heme oxygenase-1.

### 3.2. Iron Regulation in Macrophages

The key mechanism responsible for the regulation of iron homeostasis is the hepcidin-ferritin axis [46]. The target of hepcidin is FPN; this is the only iron excretion channel of macrophages. Hepcidin binds and degrades FPN, thus resulting in the accumulation of iron. Apart from the regulation of iron, hepcidin can also increase the uptake of oxidized low-density lipoprotein (ox-LDL) through CD36 and inhibit cholesterol excretion mediated by liver X receptor alpha (LXRα) and ATP-binding cassette transporterA1 (ABCA1), thus leading to a deposition of lipids inside macrophages. The deposition of ox-LDL and iron trigger the autocrine secretion of hepcidin and aggravates AS [47]. Malhotra et al. found that hepcidin deficiency was closely related with decreased iron in macrophages and AS attenuation in LDLR-/- mice [48]. Expression of the hepcidin gene is regulated by both iron and inflammation. High levels of iron stimulate the release of hepcidin via the BMP/SMAD pathway. Inflammation increases the expression of IL-6, which stimulates the phosphorylation of STAT3 that forms as a complex and increases hepcidin excretion.

In addition to the hepcidin-ferritin axis, HO-1, NCOA4, BTB domain and CNC homolog 1 (BACH1) and NEDD4-like E3 ubiquitin protein ligase (NEDD4L) can also affect iron homeostasis. HO-1 increases the amount of free iron by breaking down heme [16]. FT is important in maintaining iron homeostasis as above mentioned. NCOA4 raises the quantity of intracellular iron by attaching to FTH1 in autophagosomes and directing them to lysosomes to breakdown FT. BACH1 can increase the amount of free iron by suppressing FT genes, including both FTH and FTL [49]. NEDD4L mediates the degradation of lactotransferrin and resists ferroptosis by blocking the accumulation of iron [50].

Moreover, the expression levels of hephaestin and ceruloplasmin in AS plaques were found to be significantly lower than those in normal tissue [51]. Low levels of hephaestin and ceruloplasmin may be the underlying mechanisms for iron retention in plaques: when hephaestin and ceruloplasmin are deficient, intracellular Fe^2+^ cannot be oxidized to Fe^3+^. Thus, iron cannot be excreted out of macrophages through FPN and then iron is deposited in macrophages.

### 3.3. Iron Homeostasis in Different Subtypes of Macrophages

There are two subtypes of macrophages: M1 (pro-inflammatory macrophages) and M2 (anti-inflammatory macrophages). When induced by interferon or interferon combined with LPS, macrophages can be polarized to the M1 type. M1 macrophages secrete proinflammatory cytokines and induce the production of oxidants which promote inflammation and kill microorganisms. On the other hand, the activation of IL-4 and IL-6 can promote the differentiation of macrophages into the M2 phenotype [52]. M2 macrophages can inhibit inflammation, clear parasites and regulate immunity [53]. Studies have shown that both M1 and M2 macrophages exist in AS plaques. M1 markers are positively correlated with AS development while M2 markers are negatively correlated with AS development [54]. M2 macrophages first penetrate early stable AS plaques, but as the plaques progress, M1 macrophages gradually proliferate and become more dominant. M1 macrophages are associated with ruptured plaques or necrotic cores. This might be due to the fact that M1 scavenger receptor can phagocytose LDL without being impacted by negative feedback regulation. This causes lipid accumulation in macrophages and encourages the development of plaques.

Different types of macrophages exhibit different regulatory systems for iron homeostasis (Figure 3A,B). M1 macrophages express high levels of FT but low levels of FPN [55], HO-1 and TFR1 [56]. In contrast, M2 macrophages express low levels of FT but high levels of FPN, HO-1 and TFR1. Of these, FT stores intracellular iron; FPN is the only transport channel in macrophages that can excrete iron. HO-1 can degrade heme to obtain iron, while TFR1 mediates iron uptake in macrophages. Thus, high expression levels of FT and low expression levels of FPN leads to increased iron storage and limited iron discharge in M1 macrophages. Although low expression levels of TFR1 may reduce iron intake, M1 macrophages still express high levels of iron. This situation may be related to the fact that decreased iron intake will reduce the activity of FPN [57]. Thus, M1 macrophages tend to show high levels of iron. In contrast, M2 macrophages tend to have low levels of iron. In conclusion, different levels of iron in different types of macrophages suggest a correlation between macrophage polarization and iron metabolism.

A. In M1 macrophages, FT is highly expressed while FPN, HO-1, TFR1 and CD136 are lowly expressed. Thus, M1 macrophages show high levels of iron by decreasing iron excretion and increasing iron storage. B. In M2 macrophages, FT is lowly expressed while FPN, HO-1, TFR1 and CD136 are highly expressed. Thus, M2 macrophages show low levels of iron by increasing iron excretion and controlling iron storage. C. In M(Hb) macrophages, FPN, CD163, HO-1 and LXR are highly expressed. Thus, M(Hb) shows low levels of iron. Furthermore, LXR can stimulate the expression of ABCA1 which can increase the excretion of cholesterol. Therefore, M (Hb) are characterized by low intracellular iron and low lipid. Abbreviations: M (Hb), hemoglobin-stimulated macrophages; Fe, iron; FT, ferritin; ABCA1, ATP-binding cassette transporterA1; LXRα, liver X receptor alpha.

### 3.4. Iron Homeostasis in Special Macrophages: Hemoglobin-Stimulated Macrophages

When AS plaques rupture, intraplaque hemorrhage produces a large number of RBCs. Under an oxidative environment, RBCs tend to be broken down, thus releasing hemoglobin (Hb). Free Hb is oxidizing, pro-inflammatory and toxic. Excessive free Hb can cause damage to the normal vascular microenvironment. However, Hb and haptoglobin (Hp) can bind to form the hemoglobin-haptoglobin complex (HH) and then reduce the pro-inflammatory and oxidative effects of Hb. The HH receptor (CD163) recognizes HH and mediates the phagocytosis of HH in macrophages. This approach is also considered to be the only manner by which free Hb can be removed when plaques rupture. Macrophages that engulf large amounts of Hb are referred to as hemoglobin-stimulated macrophages (M(Hb)). Unlike M1 and M2, M(Hb) is a specific type of macrophage that is produced when plaques rupture and release RBCs. Therefore, it has been proposed that M(Hb) should be referred to as a third type of macrophage in addition to M1 and M2 macrophages [58].

M (Hb) are characterized by low intracellular iron, low levels of ROS, low levels of lipid and proinflammatory factors and increased expression levels of anti-inflammatory factors, HO-1, the mannose receptor (CD206), CD163, FPN and LXRα [59,60] (Figure 3C). High expression levels of FPN are beneficial to M (Hb) macrophages and help to expel intracellular iron and reduce iron overload. An increase in LXR can increase the outflow of iron from macrophages by decreasing hepcidin and upregulating FPN. Furthermore, LXR can also stimulate the transcription and expression of cholesterol efflux translocators (ABCA1) [61]. The up-regulation of ABCA1 can reduce lipid accumulation in macrophages by increasing cholesterol excretion. Therefore, M(Hb) macrophages are characterized by low levels of lipid and iron.

## 4. Iron Overload and Ferroptosis in Macrophages

### 4.1. Iron Overload and Ferroptosis in Macrophages Promote the Progression of AS

The appropriate amount of iron has important physiological significance for macrophages. For example, iron can affect the function and differentiation of macrophages and help to produce ROS to defend against foreign invaders [62]. However, excessive iron activity in macrophages can also have adverse effects such as promoting the progression of AS [63]. Free iron can produce large amounts of ROS via the Fenton reaction, thus causing lipid peroxidation and reducing the ratio of reduced GSH to total GSH, thus leading to an abnormal redox balance in macrophages [64]. In addition, high levels of intracellular iron can also affect the release of cytokines, increase arginase activity and reduce nitric oxide synthase activity. It can also reduce the antioxidant ability of macrophages. Macrophages with higher levels of iron can also secrete more matrix metalloproteinases (MMP) which can degrade the extracellular matrix and lead to the rupture of AS plaques. Moreover, iron overload can regulate the biological activity of macrophages by increasing the binding ability of 5-lipoxygenase to the nuclear membrane and promote inflammation in macrophages [65]. Thus, excess iron can promote the progression of AS by affecting the inflammatory response, redox balance and the release of related factors in macrophages. Therefore, a balanced intracellular iron environment can alleviate the progression of AS. In a previous study, Malhotra et al. proved that hepcidin can slow down the progression of AS by reducing the levels of iron in macrophages [48].

However, some researchers compared the effects of ammonium ferric citrate on macrophages and foam cells and found that ammonium ferric citrate could reduce the cellular activity of foam cells but had no significant effect on the activity of macrophages. Thus, these researchers concluded that iron overload induced ferroptosis in macrophage-derived foam cells but had no significant effect on the activity of macrophages [14]. From our point of view, compared with foam cells, macrophages have a more stable intracellular metabolic microenvironment. Although iron overload can affect the internal environment of macrophages, such as promoting the inflammatory response, enhancing glycolysis and inhibiting oxidative phosphorylation in the mitochondria, iron overload does not affect the cellular activity of macrophages in the short term [66]. Foam cells, in contrast, have a more fragile intracellular microenvironment than macrophages and therefore are susceptible to iron overload. Therefore, the effect of iron overload on macrophages and the progression of AS cannot be denied.

Apart from iron overload, redox imbalance and inflammation are also closely related to AS. Iron overload, lipid peroxidation and inflammatory reactions are the three main characteristics of ferroptosis. Ferroptosis is involved in the formation of foam cells from macrophages in AS. Studies have found that the expression of binding oligomerization domain 1 (NOD1) increased the levels of GPX4 and other ferroptosis regulatory proteins in macrophages and also regulated the development of AS [67]. Another study proved that macrophage ferroptosis induced by a high level of acid contributed to the development of AS plaques [68]. These results indicate the potential regulatory mechanisms underlying macrophage ferroptosis in AS. Further studies are now needed to determine the specific mechanism underlying macrophage ferroptosis in AS.

### 4.2. Iron Overload, Ferroptosis and Polarization in Macrophages

Ammonium ferric citrate can promote the expression of M1-type markers such as IL-1β, IL-6, IL-23 and TNF-α and inhibit the expression of IL 10, TNF β and CCL12 [66]. Zhou et al. found that iron overload up-regulated ROS expression by promoting p53 acetylation and increasing the activity of P300/CBP acetyltransferase; subsequently, macrophages can differentiate into the M1 type [69]. Similar results were obtained in another study using superparamagnetic iron oxide nanoparticles (SPION) for iron induction: after being processed by SPION, M2 macrophages were transformed into M1 macrophages with the up-regulation of CD86, TNF α, FT and cathepsin L [70]. Therefore, iron overload can promote the differentiation of macrophages into inflammatory types, thus promoting the inflammation response in AS plaques and accelerating the progression of AS. In contrast, the treatment of macrophages with iron-reducing methods such as iron chelating agents, iron inhibitors, an iron restricted diet or hepcidin reduction can promote the differentiation of macrophages into the M2 phenotype. Malhotra et al. [48] observed a significant reduction in the number of M1 macrophages in hepcidin-deficient mice when compared with non-hepcidin-deficient controls.

However, studies have also shown that an iron-deficient diet worsened the inflammatory response while an iron-rich diet raised the expression levels of the M2 markers Arg1 and Ym1 [71]. This means that an excess of iron may cause macrophages to polarize to the M2 type, thus leading to the opposite conclusion. We hypothesize that reactive iron, not overall iron stores, is what initiates inflammatory responses. Increased iron loading in a healthy body may encourage macrophage differentiation into the M2 phenotype and thus facilitate the body’s anti-inflammatory response. However, an increase in active iron in macrophages, particularly in pathological situations such as AS, may disrupt the already fragile iron homeostasis and cause the M1 phenotype of inflammatory macrophages to differentiate, thus exacerbating the inflammatory response.

In addition to iron overload increasing the number of M1 macrophages, studies have found that exposure to LPS and interferon-Y (IFN-Y) also increases the number of M1 macrophages [72]. Furthermore, the simultaneous exposure of macrophages to ox-LDL and LPS/IFN-γ produces a mixed macrophage phenotype called the Mox/M1 phenotype which closely resembles the M1 phenotype [73]. Iron overload and lipid peroxidation are the key mechanisms underlying ferroptosis. Thus, ferroptosis may have an important effect on the polarization of macrophages. Researchers previously used genetic testing to show that suppressor of cytokine signaling-1 (an inducer of ferroptosis) was associated with M1 macrophages and that FTN1 (an inhibitor of ferroptosis) was associated with M2 macrophages [74]. Tang et al. [75] found that the ferroptosis regulator ribonucleotide reductase small subunit M2 (RRM2) was closely related to macrophage infiltration and polarization; by knocking out RRM2, M2 markers were inhibited and M1 markers were induced. These studies suggest that ferroptosis is mostly accompanied by M1 macrophages which is consistent with the effect of iron overload on macrophage polarization. Moreover, M1 and M2 macrophages not only show different states of iron metabolism but also show different sensitivities to ferroptosis. M1 macrophages are more resistant to ferroptosis than M2 macrophages. This phenomenon may be related to iNOS/NO• as M1 macrophages express higher levels. Researchers have also found that iNOS/NO• can inhibit ferroptosis induced by RSL and proposed that iNOS/NO• may even replace GPX4 in the ferroptosis defense mechanism [76].

Hu et al. [66] measured the extracellular acidification rate (ECAR) of macrophages and found that the ECAR of iron-induced macrophages was increased, along with glycolysis-related RNA and proteins. This suggests that iron overload may affect the polarization of macrophages through glycolysis. Another study aggravated the pressure of ferroptosis in macrophages using a nanocrystal preparation of iron (MIL88B/RSL3) and found that macrophages in the MIL88B/RSL3 group exhibited an increased glycolysis reserve, glycolysis capacity, and glycolysis level, but decreased basal mitochondrial respiration, maximum respiration, residual respiration capacity and ATP production. Under the pressure of ferroptosis, M2 macrophages were induced to differentiate into M1 macrophages [77]. These results suggest that iron overload and ferroptosis may promote the polarization of macrophages to the M1 phenotype via enhanced glycolysis.

### 4.3. Iron Metabolism and Ferroptosis in M (Hb) Type Macrophages

The characteristics of M (Hb) macrophages include low iron, low ROS, low inflammatory factors and high expression levels of ABCA1 which increase lipid excretion. According to the characteristics of M (Hb), it appears that M (Hb) may be a protective factor for plaques in advanced AS. However, other studies pointed out that low levels of iron in M (Hb) may promote the progression of AS in some respects. Low iron levels in M(Hb) inhibits iron-dependent prolyl hydroxylase protein (PHDs), and low PHDs activate hypoxia-inducible factor 1α (HIF-1α) and promotes the expression of vascular endothelial growth factor (VEGF). VEGF mediates the expression of vascular endothelial target genes, increases the permeability of endothelial cells, promotes angiogenesis, and induces the expression of inflammatory factors [78]. During the progression of AS, hepcidin can inhibit the development of plaques by reducing iron in macrophages and controlling the inflammatory response. However, in areas of plaque hemorrhage in advanced AS, low levels of iron causes a reduction in iron-dependent PHDs, which in turn leads to an inflammatory response and endothelial dysfunction. The effect of hepcidin may actually be deleterious at this stage [79]. Thus, maintaining low levels of iron may be protective during the progression of AS, but low levels of iron may also promote the progression of AS in areas affected by plaque hemorrhage.

Ferroptosis may occur in M (Hb) macrophages despite the presence of low iron content. Youssef et al. found that following the phagocytosis of a large number of aged or damaged RBCs, macrophages showed increased levels of ROS, enhanced lipid peroxidation and increased expression of the PTGS2 gene. Ferroptosis occurs in macrophages and leads to a decline in numbers. Additionally, these changes can be reversed by Fer-1 [80]. Another study proved that after phagocytosis of RBCs, cell death of M2 and M0 increased, not M1. Additionally, the enhanced lipid peroxidation and macrophage cell death brought on by RBCs can be reversed by fer-1 or GPX4 overexpression. However, lipid peroxidation and cell death were not prevented by inhibitors of other types of cell death. These results confirmed that macrophage death following RBC phagocytosis may arise from ferroptosis [81].

## 5. Prospect

In this paper, iron homeostasis, iron regulation and iron overload in macrophages and their correlations with AS were described in detail. The impact of iron on AS is not simple. While an increase in active iron may accelerate AS by encouraging ferroptosis, an increase in total iron storage may prevent the progression of AS. Additionally, low iron M2 are strongly linked to AS remission, but low iron M (Hb) may hasten AS development by enhancing endothelial cell permeability and inflammatory factor expression. Clinical studies also proved that the concentration of iron ions in patients’ peripheral blood was closely related to AS [82]. Due to the complexity of iron’s effect on AS, understanding the role that iron plays in AS can help guide clinical therapy. We hope that this review will offer new insights into the relationship between AS and iron in macrophages.

Ferroptosis, one of the trendiest topics in current study, has been proved to have an impact on a variety of diseases. Drugs targeting of ferroptosis may have potential clinical application value. However, most of which are still in the experimental stage, with some drawbacks such as long treatment cycles, difficult dosage management and unknown treatment effects. DFO, for example, may cause neurotoxicity at high doses [83]. AS is a disease with a complicated pathophysiology and no particular medications. We investigated the connection between ferroptosis and AS through mechanisms, related molecules and genes. We also investigated the connection between iron overload and ferroptosis in macrophages and AS from the viewpoints of iron homeostasis, macrophages polarization and special types of macrophages. Our review may provide references and tips for further research. We believe that as more in-depth study into ferroptosis is conducted, the development of therapeutic medicines based on it is expected to bring considerable benefit to the future treatment of AS.

## 6. Conclusions

This review describes the basic concept of ferroptosis, a newly discovered form of cell death caused by iron-dependent redox imbalance. The three main pathways against ferroptosis are the GPX4-GSH, FSP1-CoQ10 and GCH1-BH4 axes. Evidence indicates that ferroptosis plays an important role in the occurrence and development of AS. In this paper, we first focused on macrophages, which have rarely been studied from the perspective of ferroptosis. Then, we focused on the effects of macrophage iron overload and ferroptosis on the progression of AS by specifically considering iron metabolism. Iron overload and ferroptosis have similar effects on polarization in macrophages and the progression of AS. In addition, macrophage death following RBC phagocytosis may arise from ferroptosis and M(Hb) can promote the development of AS. However, there are still few studies focusing on the effect of macrophage ferroptosis and AS. Furthermore, the exact mechanisms responsible for how macrophage ferroptosis affects AS remain unknown. Therefore, further studies are strongly recommended to confirm the specific relationship between macrophage ferroptosis and AS.

## Figures and Tables

**Figure 1 biomolecules-12-01702-f001:**
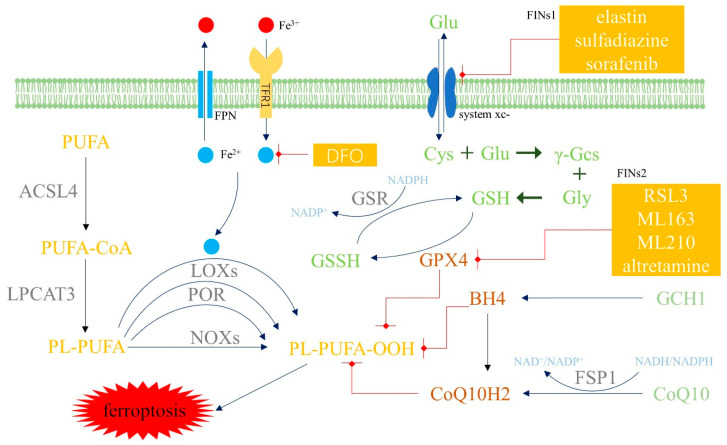
The pathway of ferroptosis.

**Figure 2 biomolecules-12-01702-f002:**
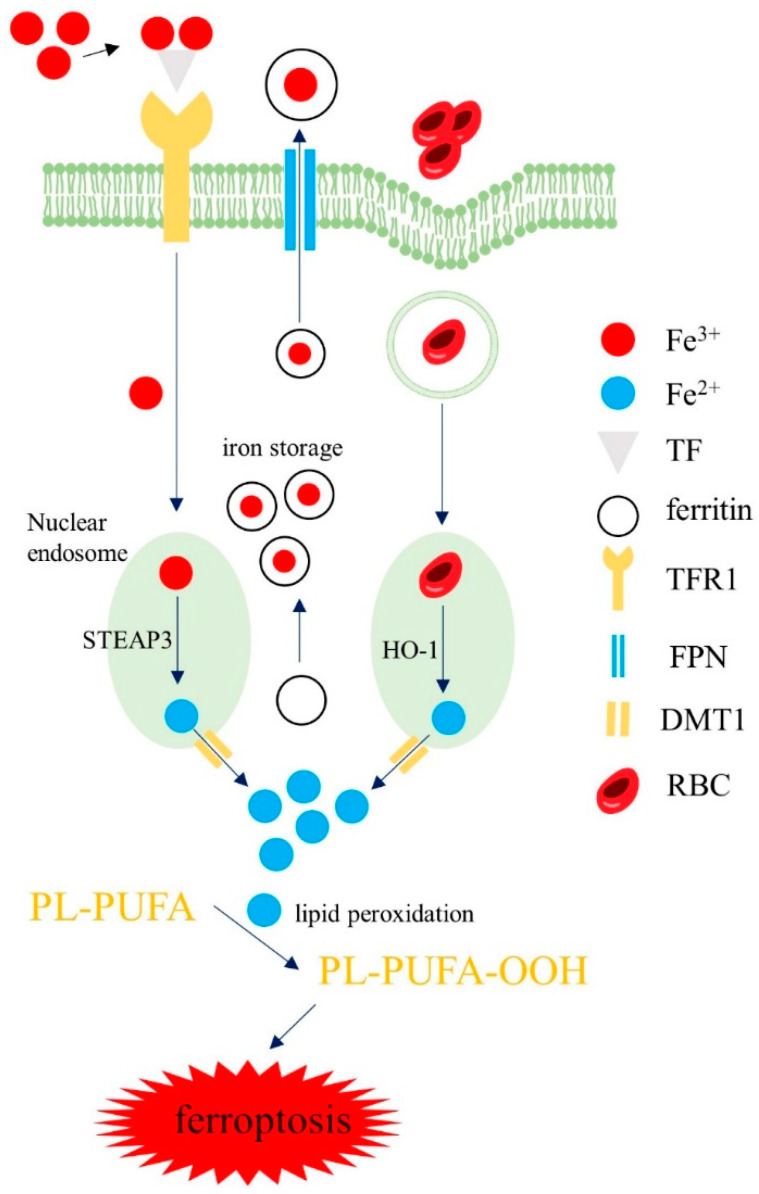
Iron homeostasis in macrophages.

**Figure 3 biomolecules-12-01702-f003:**
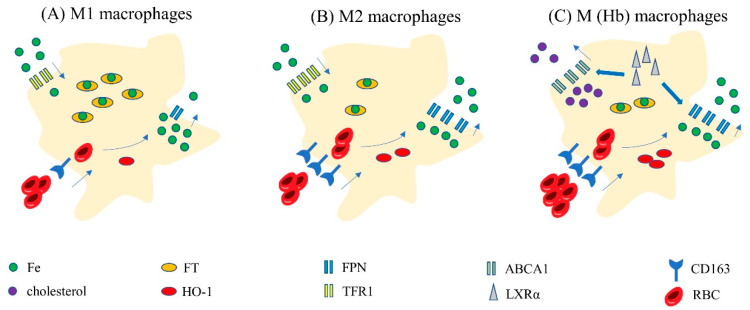
Characteristics of iron metabolism in M1, M2 and M (Hb) macrophages. (**A**) M1 macrophages; (**B**) M2 macrophages; (**C**) M(Hb) macrophages.

**Table 1 biomolecules-12-01702-t001:** Effects of different forms of macrophage death on atherosclerosis.

Forms of Death	Influence	Reference
Apoptosis	Protective in early stage and harmful in late stage	[6]
Pyroptosis	Enlarges necrotic core and destroys plaque stability	[5]
Necroptosis	Enlarges necrotic core and induces inflammation	[7]
Autophagy	Reduces ROS and plays a protective role in AS	[8]

ROS: reactive oxygen species; AS: atherosclerosis.

**Table 2 biomolecules-12-01702-t002:** Inducers and inhibitors of ferroptosis associated with iron metabolism.

	Regulators of Ferroptosis	Effects	Subjects	Reference
Inducers	Ammonium ferric citrate	Increases intracellular iron load	Foam cells	[14]
	NCOA4	Increases intracellular iron storage and degrades ferritin	Macrophages	[15]
	BAY 11-7085	Up-regulates HO-1 through Nrf2-SLC7A11-HO-1 pathway and increases free iron	Cancer cells	[16]
	HO-1 and Hb	Increases free iron by decomposing Hb	Cancer cells	[16]
	WA	Decreases the activity of GPX4; activates HO-1 by KEAP1	Neuroblastoma cells	[17]
	Siramesine and lapatinib	Increase transferrin and iron entering cells; decrease FPN and iron leaving cells	Breast cancer cells	[18]
	CaFe@DMSN/C NPs	Increases intracellular iron and ROS	4T1 and Raw264.7 cells	[19]
Inhibitors	Fer-1	Decreases intracellular iron and ROS	MAECs	[20]
	Curcumin and EGCG	Chelate iron and limit the accumulation of iron in cells	Mouse MIN6 pancreatic-cells	[21]
	Heat shock protein 27	Restricts LIP; Reduces iron intake through TfR1	Chinese hamster lung fibroblast cells	[22]
	CISD1,2	Decreases iron and ROS in mitochondria	Human HCC cell lines/ HNC and SNU cell lines	[23,24]
	DFO	Chelates iron and limits iron accumulation in cells	HT-1080	[10]

NCOA4: nuclear receptor coactivator 4; HO-1: heme oxygenase-1; Hb: hemoglobin; WA: withaferin A; KEAP1: kelch-like ECH-associated protein 1; FPN: ferroportin; Fer-1: ferrostatin-1; MAECs: mouse aortic endothelial cells; EGCG: (—)-epigallocatechin-3-gallate; LIP: labile iron pools; TFR1: transferrin receptor 1; CISD1: CDGSH iron sulfur domain 1; CISD2: CDGSH iron sulfur domain 2; DFO: deferoxamine.

## Data Availability

Not applicable.

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
