# Peer review of "The Role of Macrophage Iron Overload and Ferroptosis in Atherosclerosis"

_biomolecules, 2022, doi:10.3390/biom12111702_

Round 1

Reviewer 1 Report

In this review the authors summarize the role of macrophage iron overload and ferroptosis in the pathogenesis of atherosclerosis in a very comprehensive way. The authors provide a general overview of the mechanism of ferroptosis, and the iron homeostasis of macrophages, also in relation with different macrophage polarization phenotypes. Then, they highlight the existing connection between macrophage iron overload and ferroptosis and atherosclerosis development. All topics are covered in detail and I have no concerns about them. I found the review well written and clear. I have no other suggestions.

Reviewer 2 Report

Thank you for submitting this manuscript for review. This is a well written narrative review on the role of macrophage / iron homeostasis and atherosclerosis that describes the cellular mechanisms relevant to these processes and highlights areas of further research. 

Minor typos are present in the manuscript. Overall comprehensive. 

I appreciate this is not a systematic review but information on the search strategy would be helpful. 

I would suggest inserting a section on the translational relevance of what has been presented in this narrative review and how this can impact upon patient practice. 
